# Multiple decay events target *HAC1* mRNA during splicing to regulate the unfolded protein response

**Patrick D Cherry[1,2], Sally E Peach[1], Jay R Hesselberth[1]\***

[1]Department of Biochemistry and Molecular Genetics, Program in Molecular Biology, School of Medicine, University of Colorado, Aurora, United States; [2]RNA Bioscience Initiative, School of Medicine, University of Colorado, Aurora, United States

**Abstract** In the unfolded protein response (UPR), stress in the endoplasmic reticulum (ER) activates a large transcriptional program to increase ER folding capacity. During the budding yeast UPR, Ire1 excises an intron from the *HAC1* mRNA and the exon products of cleavage are ligated, and the translated protein induces hundreds of stress-response genes. Using cells with mutations in RNA repair and decay enzymes, we show that phosphorylation of two different *HAC1* splicing intermediates is required for their degradation by the 5′→3′ exonuclease Xrn1 to enact opposing effects on the UPR. We also found that ligated but 2′-phosphorylated *HAC1* mRNA is cleaved, yielding a decay intermediate with both 5′- and 2′-phosphates at its 5′-end that inhibit 5′→3′ decay and suggesting that Ire1 degrades incompletely processed *HAC1*. These decay events expand the scope of RNA-based regulation in the budding yeast UPR and have implications for the control of the metazoan UPR.

DOI: https://doi.org/10.7554/eLife.42262.001

## Introduction

During the unfolded protein response (UPR), protein folding stress in the lumen of the endoplasmic reticulum leads to oligomerization of the transmembrane kinase/endoribonuclease Ire1 and the processing of a cytoplasmic mRNA to yield splicing intermediates with 2′,3′-cyclic phosphate ($PO_4$) and 5′-hydroxyl (OH) termini (*Gonzalez et al., 1999*). In budding yeast, excision of an intron from the *HAC1*[u] mRNA ('u' denoting the unspliced mRNA) by Ire1 is followed by exon ligation by the multifunctional Trl1 RNA ligase (*Sidrauski et al., 1996*) involving 5′-phosphorylation of the 5′-OH product, adenylylation of the 5′-$PO_4$, and resolution of the 2′,3′-cyclic $PO_4$ to a 2′-$PO_4$/3′-OH. The newly produced 3′-OH serves as the nucleophile to attack the 5′-adenylate intermediate, yielding a ligated mRNA with an internal 2′-$PO_4$. The 2′-$PO_4$ is assumed to be removed in a separate reaction by the 2′-phosphotransferase, Tpt1, in a $NAD^+$-dependent reaction (*Culver et al., 1997*). The spliced mRNA, called *HAC1*[s] mRNA ('s' denoting spliced mRNA) (*Li et al., 2018*), is translated into a transcription factor that activates the expression of dozens of stress-response genes to mitigate protein-folding stress (*Ron and Walter, 2007*). In addition, Hac1 activates its own promoter in a positive feedback loop that generates more *HAC1*[u] and permits sustained UPR activation (*Ogawa and Mori, 2004*) (*Figure 1A*).

Control of this positive feedback loop ensures UPR suppression during normal growth and rapid activation upon stress exposure. To facilitate the control of UPR activation, *HAC1*[u] contains *cis*-regulatory elements that suppress unintended translation and promote rapid processing. A long-range base-pairing interaction between the 5′-UTR and intron prevents ribosome initiation to suppress translation of *HAC1*[u] mRNA (*Chapman and Walter, 1997*; *Di Santo et al., 2016*). If a ribosome

**\*For correspondence:**
jay.hesselberth@gmail.com

**Competing interests:** The authors declare that no competing interests exist.

**eLife digest** Like any economical factory, cells tune the size of their protein assembly line to suit demand. Proteins consist of strings of amino acids, built from template molecules called mRNAs, that must be folded into specific 3D structures for them to work correctly. If these protein strings are produced faster than they can be folded, the cell triggers the unfolded protein response. This response slows protein production, gets rid of any misshapen proteins, and increases the size of the protein assembly line.

It is not clear exactly how the unfolded protein response is tuned, though an mRNA molecule called *HAC1* is known to signal the response. First, enzymes remove a short section of *HAC1* and join the remaining parts back together in a process called splicing. Spliced *HAC1* is then used as a template to make a protein that activates the unfolded protein response.

To understand more about this processing of *HAC1*, Cherry et al. studied yeast cells that had mutated, non-working versions of some of the enzymes that repair and degrade RNA. This revealed that the splicing of *HAC1* competes with another process that breaks down mRNA. Under normal conditions, this means that *HAC1* is degraded before it can trigger the unfolded protein response. In addition, for the cell to trigger the unfolded protein response, it needs to break down the part of *HAC1* that is removed during splicing. Otherwise, the removed section interferes with the spliced *HAC1* mRNA, preventing it from being a signal to activate the unfolded protein response.

Cherry et al. also found that a unique, chemically modified fragment of *HAC1* mRNA was protected from degradation. They do not know how the unique chemical modification regulates the unfolded protein response, but stabilizing modifications are generally useful in RNA biology.

Understanding how the unfolded protein response is tuned could help researchers to find new ways to treat conditions where it does not work correctly, such as neurodegeneration, diabetes and cancer. Additionally, researchers are already trying to develop treatments for a number of diseases that work by inserting new RNA molecules into cells. Understanding how the chemical modification discovered by Cherry et al. protects RNAs from degradation could therefore improve the effectiveness of such treatments.

DOI: https://doi.org/10.7554/eLife.42262.002

initiates on $HAC1^u$, translation through the 5′-exon/intron junction yields a truncated protein with an intron-encoded C-terminal peptide 'degron' that targets it for ubiquitylation and degradation (*Di Santo et al., 2016*). A stem-loop (the '3′-BE') in the 3′-untranslated region of *HAC1* tethers the mRNA to the ER membrane, ensuring rapid Ire1-mediated cleavage following ER stress (*Aragón et al., 2009*).

Previous work found unexpected roles for RNA decay and repair enzymes acting on *HAC1* mRNA in the budding yeast unfolded protein response. Ire1 is a metal-ion-independent endonuclease that produces RNA cleavage products with 5′-OH termini (*Gonzalez et al., 1999*). In cells lacking the cytoplasmic 5′→3′ exonuclease Xrn1, *HAC1* splicing intermediates accumulate with 5′-$PO_4$ termini, indicating that a RNA 5′-kinase phosphorylates *HAC1* processing intermediates and that not all *HAC1* splicing intermediates are productively ligated (*Harigaya and Parker, 2012*; *Peach et al., 2015*). In addition to its role in *HAC1* exon ligation, Trl1 is required to relieve translational attenuation of $HAC1^s$ by an unknown mechanism (*Mori et al., 2010*). In cells expressing the T4 bacteriophage RNA repair enzymes PNK and RNL1 in lieu of *TRL1*, ligated *HAC1* molecules contained single nucleotide deletions from the 3′-terminus of the 5′-exon, indicating that a 3′→5′ exonucleolytic activity acts on the cleaved 5′-exon (*Schwer et al., 2004*) and nuclear 3′→5′ decay of $HAC1^u$ liberates the 3′-BE, tuning the activation potential of the UPR (*Sarkar et al., 2018*).

Recent studies showed that RNA decay also plays a role in the UPR in other organisms. During UPR activation in the fission yeast, Ire1 incises specific mRNAs to promote their stabilization or degradation (*Guydosh et al., 2017*; *Kimmig et al., 2012*). This mode of Ire1 cleavage is similar to the metazoan Regulated Ire1-Dependent Decay (RIDD) pathway wherein Ire1 incises some ER-localized mRNAs and the cleavage products are degraded by Xrn1 and the cytoplasmic exosome (*Hollien and Weissman, 2006*).

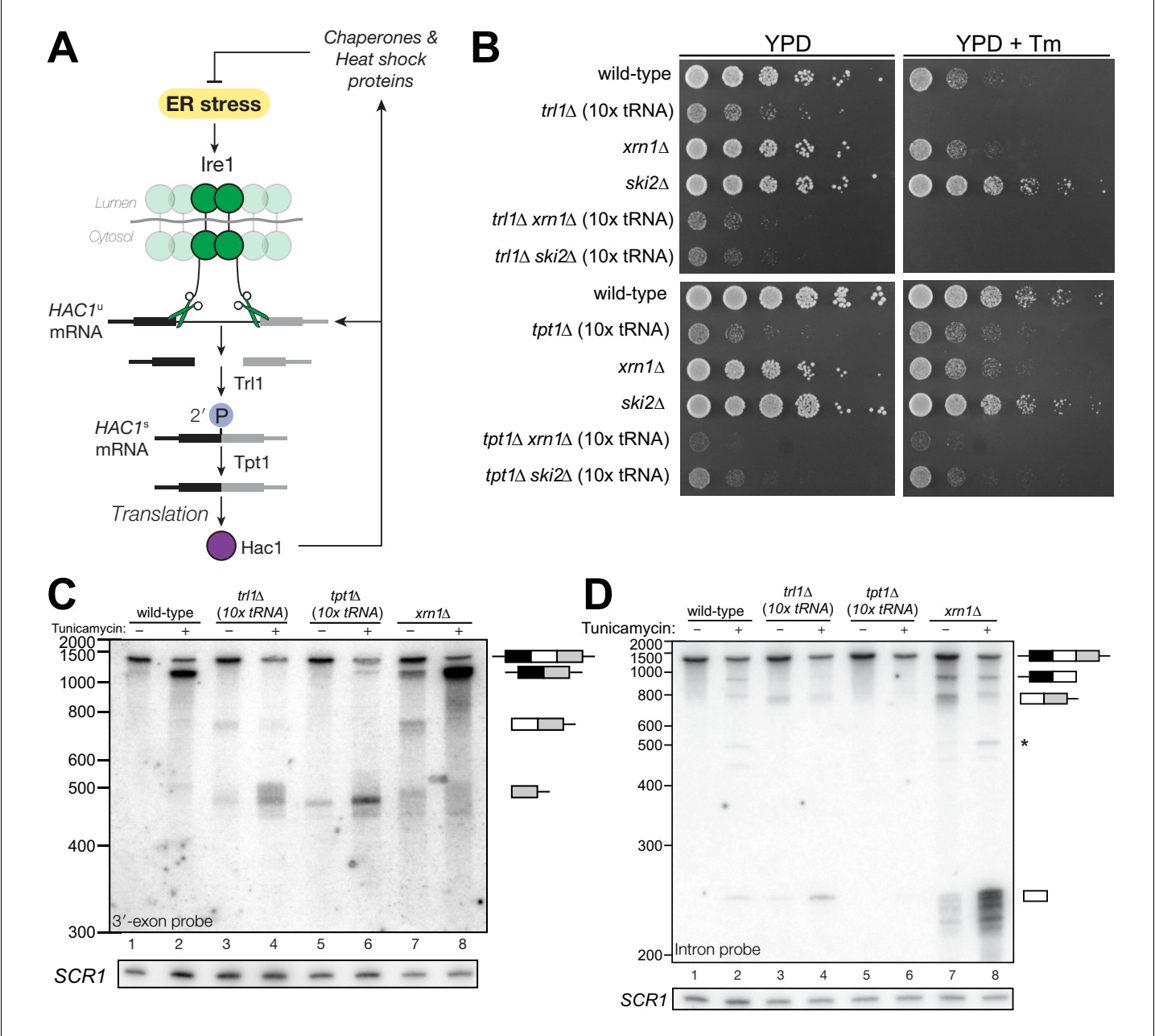

**Figure 1.** *HAC1* mRNA processing defects in RNA repair and decay mutants. (**A**) Schematic of the budding yeast unfolded protein response. ER stress activates Ire1 (green), which excises an intron (thin line) from *HAC1*[u] mRNA. The 5′ (black) and 3′ (grey) exons are ligated by Trl1 yielding spliced *HAC1* (*HAC1*[s]) with a 2′-phosphate at the newly-formed ligation junction, which is subsequently removed by the 2′-phosphotransferase Tpt1. *HAC1*[s] mRNA is translated into a transcription factor (Hac1, purple) that upregulates the *HAC1* gene itself (a positive feedback loop), as well as several chaperones and heat shock proteins that resolve the stress. (**B**) Yeast cells with mutations in RNA repair and decay factors were serially diluted (5-fold) and spotted onto agar media (YPD and YPD containing tunicamycin (Tm; 0.16 μg/mL)), grown at 30℃ for 2 days, and photographed. The '10x tRNA' plasmid encodes 10 intronless tRNAs that bypass the lethality of *trl1Δ* and *tpt1Δ* (*Cherry et al., 2018*). The top panels depict cells with deletions of the RNA ligase *TRL1* and the bottom panels depict growth of cells deletions of the 2′-phosphotransferase *TPT1*. (**C**) *HAC1* processing in RNA repair and decay mutants (3′-exon probe). *HAC1*[u] cleavage and ligation were analyzed in mutants of *TRL1* RNA ligase and *TPT1* 2′-phosphotransferase by denaturing acrylamide gel northern blotting using a probe to the *HAC1* 3′-exon. Diagrams of *HAC1*[u], *HAC1*[s], and *HAC1* splicing intermediates are drawn next to predominant bands (see *Table 1* for descriptions and sizes of all annotations). *HAC1*[u] is cleaved and ligated to produce *HAC1*[s] in wild-type cells (lanes 1 and 2). Intron/3′-exon and 3′-exon splicing intermediates accumulate in *trl1Δ* cells, but *HAC1*[s] is not produced (lanes 3 and 4). In *tpt1Δ* cells, a small amount of cleaved 3′-exon is present in the absence of tunicamycin (lane 5), whereas *HAC1*[s] and 3′-exon accumulate upon tunicamycin induction (lane 6). Cells lacking *xrn1Δ* grown in the absence of tunicamycin produce *HAC1*[s] and Intron/3′-exon and 3′-exon splicing intermediates (lane 7), and tunicamycin addition causes an increase in production of *HAC1*[s] (lane 8). The blot was stripped and reprobed using a probe for *SCR1* as a loading control. (**D**)
*Figure 1 continued on next page*

*Figure 1 continued*

*HAC1* processing in RNA repair and decay mutants (intron probe). Linear intron (252 nt) is excised from *HAC1*ᵘ upon tunicamycin treatment (lanes 1 and 2) and linear intron is excised and accumulates in *trl1Δ* cells in the presence and absence of treatment (lanes 3 and 4). Excised intron is present a low levels in *tpt1Δ* cells (lanes 5 and 6), whereas *xrn1Δ* cells accumulate high levels of full-length intron and shorter, intron-derived decay intermediates (lanes 7 and 8). A star denotes excised and circularized intron, which migrates at ~500 nt.

DOI: https://doi.org/10.7554/eLife.42262.003

Here, we used budding yeast with mutations in RNA repair and decay enzymes to show that *HAC1* splicing intermediates are processed at multiple steps prior to ligation, limiting the impact of spurious Ire1 activation and unintentional *HAC1* cleavage. Our studies also show that incompletely spliced *HAC1*ˢ mRNA is targeted for degradation, which may be used to attenuate the UPR.

## Results

### RNA repair mutants have unique *HAC1* mRNA processing defects

We recently showed that the functions of the essential RNA repair enzymes Trl1 and Tpt1 in budding yeast can be genetically bypassed by the expression of intronless tRNAs, which are able to support translation in *trl1Δ* and *tpt1Δ* cells (*Cherry et al., 2018*). Because Trl1 is required for *HAC1*ˢ ligation and subsequent UPR activation (*Sidrauski et al., 1996*), *trl1Δ* cells are unable to grow on media containing tunicamycin (*Figure 1B*). In contrast, the general growth defect of *tpt1Δ* cells is unaffected by tunicamycin (*Cherry et al., 2018*) (*Figure 1B*), indicating that these cells can activate the UPR. Combination of *trl1Δ and tpt1Δ* with mutations in 5′→3′ and 3′→5′ decay factors *xrn1Δ* or *ski2Δ* led to more pronounced growth defects than the single deletions, but removal of these decay factors did not affect the growth deficit of *trl1Δ* or *tpt1Δ* cells on tunicamycin (*Figure 1B*).

Given the multiple enzymatic roles of Trl1 and Tpt1 during RNA repair, we sought to understand how the loss of these enzymes affected *HAC1* mRNA splicing. We visualized *HAC1* splicing intermediates by northern blotting with probes for the *HAC1* 3′-exon and intron (*Figure 1C,D*) and found cleavage and ligation of *HAC1*ᵘ in wild-type cells in the presence of tunicamycin, leading to high levels of *HAC1*ˢ. As expected, *trl1Δ* cells lacking RNA ligase activity did not produce *HAC1*ˢ upon tunicamycin treatment. However, cleaved 3′-exon and intron accumulated upon tunicamycin treatment in *trl1Δ* cells (*Figure 1C,D*), indicating a defect in 3′-exon decay. Cleaved *HAC1* 3′-exon often appears as a smear of products between ~450 nt and ~575 nt (*Figure 1C*); we attribute this size

**Table 1.** *HAC1* processing intermediates.

| Name | Size (nt) | Visual summary | Description |
|---|---|---|---|
| *HAC1*ᵘ | 1450* |  | Full-length, genomic *HAC1* transcript |
| *HAC1*ˢ | 1198* |  | Spliced *HAC1*; intron removed |
| HAC1 5′-exon | 728 |  | Everything 5′ of the intron |
| Cleaved 5′-exon | ~678 |  | Fragment of 5′-Exon missing ~ 50 nt off its 3′-end |
| HAC1 intron | 252 |  | Liberated intron (alone) |
| circularized intron | ~500 |  | Circularized intron, visible in wild-type and RtcB cells |
| HAC1 3′-exon | 474* |  | Everything 3′ of the intron |
| Cleaved 3′-exon | ~524* |  | 3′-Exon with ~ 50 nt of 5′-Exon on its 5′-end |
| 5′-exon + intron | 980 |  | 5′-exon + Intron |
| Intron + 3′-exon | 726* |  | Intron + 3′-exon |

Sizes of *HAC1* processing intermediates are predicted from strand-specific RNA sequencing data (*Levin et al., 2010*) mapped to the sacCer1 genome.
*Size does not include poly(A) tail.

DOI: https://doi.org/10.7554/eLife.42262.004

heterogeneity to differences in poly(A) tail presence or length, as the 5′-ends of these products occur uniformly at one site (Figure 5A).

The 2′-phosphotransferase Tpt1 is essential in budding yeast to remove 2′-phosphate groups from ligated tRNAs (*Culver et al., 1997*), but its role in *HAC1* mRNA processing during the UPR has not been defined. Although the growth of *tpt1Δ* cells is unaffected by tunicamycin (*Figure 1B*), specific perturbations of *HAC1* processing in *tpt1Δ* cells indicate that residual 2′-phosphate groups on *HAC1* mRNA cause defects in cleavage and ligation. Whereas tunicamycin treatment led to cleavage of *HAC1*ᵘ and production of *HAC1*ˢ in *tpt1Δ* cells, the levels of *HAC1*ˢ (*Figure 1C*) and excised intron (*Figure 1D*) are lower than in wild-type cells. In addition, despite the fact that *tpt1Δ* cells have functional RNA ligase, cleaved 3′-exon accumulated to high levels upon tunicamycin treatment (*Figure 1C*, lane 6).

## Kinase-mediated decay of cleaved *HAC1* 3′-exon competes with its ligation

To further investigate the 3′-exon decay defect, we examined splicing of *HAC1* in *xrn1Δ* cells. We found that *HAC1*ˢ accumulated in the absence of tunicamycin (*Figures 1C*, *2A, D, E and F*). This promiscuous processing was surprising given that *HAC1*ˢ is undetectable in wild-type cells under normal growth conditions, and it suggested that Xrn1 somehow limits production of *HAC1*ˢ. In *xrn1Δ* cells, 3′-exon accumulated to modest levels in both the absence and presence of tunicamycin (*Figure 2A*), whereas in *trl1Δ* cells, *HAC1* 3′-exon accumulated to higher levels (*Figure 2A and B*). Moreover, the abundance of 3′-exon was similar in *trl1Δ* and *trl1Δ xrn1Δ* cells (*Figure 2A*), indicating that Xrn1 requires Trl1 for 3′-exon degradation. Previous work showed that Trl1 5′-kinase activity is required for the Xrn1-mediated degradation of excised tRNA introns in budding yeast (*Wu and Hopper, 2014*), and we considered whether this pathway also degraded *HAC1* 3′-exon. Indeed, expression of a kinase-inactive version of Trl1 (Trl1-D425N) (*Wang et al., 2006*) did not restore Xrn1-mediated decay of the 3′-exon (*Figure 2B*), affirming that the 5′-kinase activity of Trl1 ligase is required for Xrn1-mediated suppression of *HAC1* splicing. We also tested whether the ligase activity of Trl1 affected *HAC1* 3′-exon abundance using an adenylyl-transferase/ligase defective allele (Trl1-K114A) (*Sawaya et al., 2003*) and found that additional 3′-exon accumulates compared to wild-type (*Figure 2C*), indicating that ligation also contributes to processing of free 3′-exon. Furthermore, we examined the accumulation of 3′-exon in cells lacking Dxo1, a distributive, 5′-phosphate-dependent 5′→3′ exonuclease (*Chang et al., 2012*), and found that *HAC1* 3′-exon accumulation was unaffected in *dxo1Δ* cells. In addition, the levels of 3′-exon were similar in *xrn1Δ* and *dxo1Δ xrn1Δ* cells (*Figure 2D*), indicating that Xrn1 is the primary factor responsible for 5′→3′ decay of the 3′-exon.

Together these data indicate that ligation and Xrn1-mediated 5′→3′ decay compete for the 5′-phosphorylated 3′-exon splicing intermediate (*Figure 2G*, top). Examination of *HAC1* splicing in *trl1Δ* cells expressing the *E. coli* RtcB RNA ligase (*Tanaka et al., 2011*) provided additional evidence of a competition between ligation and decay. RtcB catalyzes ligation of 2′,3′-cyclic $PO_4$ and 5′-OH RNA termini via a unique mechanism involving nucleophilic attack of the 5′-OH on a 3′-guanylate intermediate; accordingly, RtcB does not have 5′-kinase activity (*Chakravarty et al., 2012*). We found that upon tunicamycin treatment, *HAC1*ˢ was produced in *trl1Δ* (*RtcB*) cells (*Figure 2E*), as shown previously (*Tanaka et al., 2011*). However, under normal growth conditions, *trl1Δ* (*RtcB*) cells also promiscuously spliced *HAC1*ˢ at levels similar to *xrn1Δ* cells (*Figure 2E and F*). We propose that because *HAC1* ligation by RtcB does not involve a 5′-phosphate intermediate, Xrn1 is unable to degrade the 5′-hydroxyl exon product of Ire1 cleavage, tipping the balance toward ligation and producing *HAC1*ˢ under normal growth conditions (*Figure 2G*, bottom). Thus Xrn1-mediated decay of *HAC1* 3′-exon appears to counteract a low rate of background Ire1 cleavage to ensure the UPR is only activated when legitimately stressed.

## Kinase-mediated decay of excised intron is required for *HAC1*ˢ translation

In several instances, cells with mutations in repair and decay factors can splice *HAC1* but fail to grow on media containing tunicamycin (*Figure 3A*), indicating that *HAC1*ˢ production is not sufficient to activate the UPR. We assayed expression of *KAR2*, an ER chaperone and direct target of the Hac1 transcription factor (*Kohno et al., 1993*), and found that all repair and decay mutants express

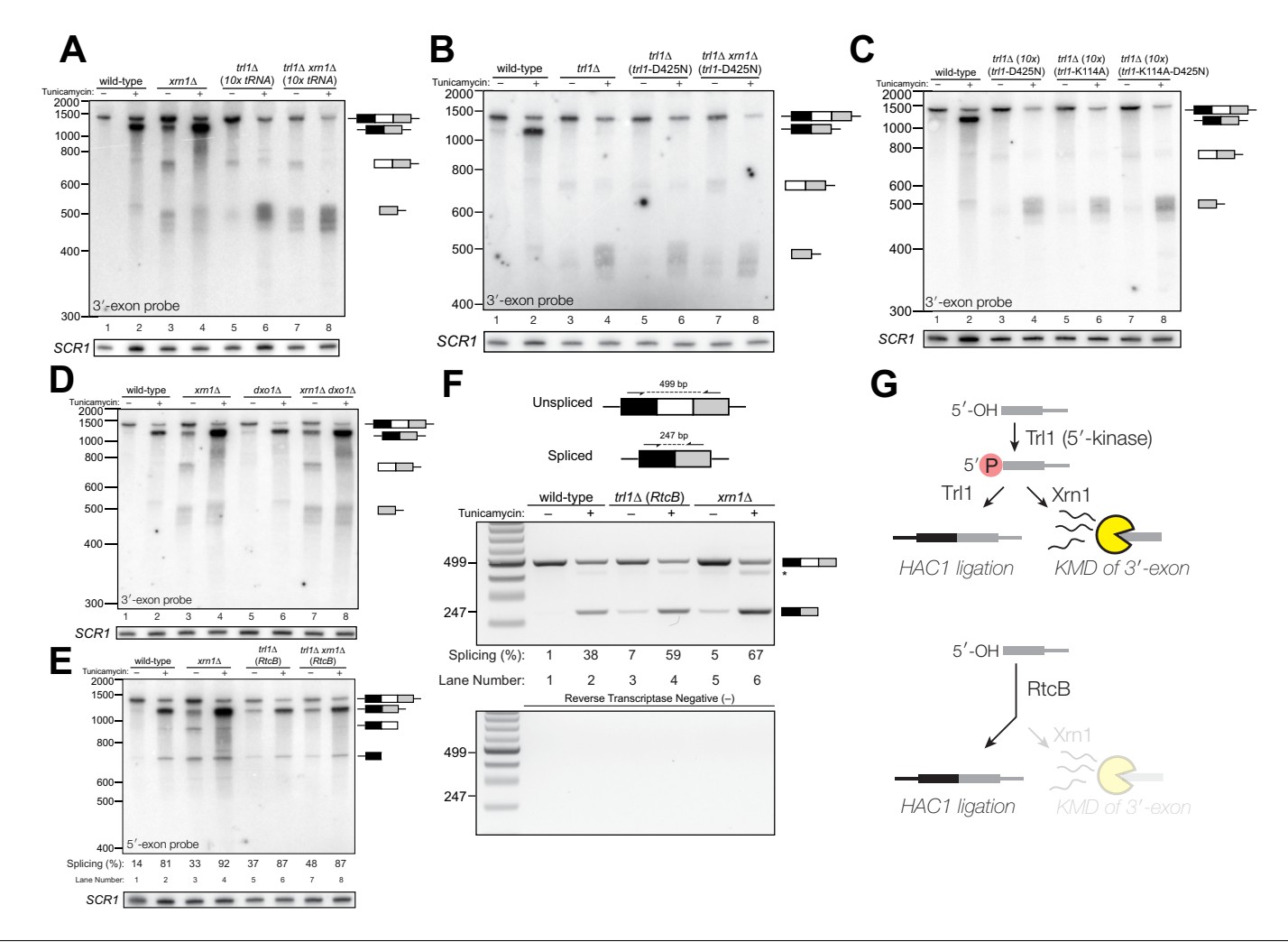

**Figure 2.** Kinase-mediated decay of *HAC1* 3'-exon competes with its ligation. (**A**) Decay of cleaved *HAC1* 3'-exon requires Trl1. A northern blot for *HAC1* 3'-exon reveals that *xrn1Δ* (lanes 3 and 4) and *trl1Δ* (lanes 5 and 6) are mutations sufficient to cause 3'-exon to accumulate compared to wild-type. In *xrn1Δ* cells, the accumulation appears tunicamycin-treatment independent, whereas in *trl1Δ* cells the accumulation increases upon treatment. The accumulation is also present in *xrn1Δ trl1Δ* cells (lanes 7 and 8). (**B**) Decay of cleaved *HAC1* 3'-exon requires the catalytic activity of Trl1 5'-kinase. We expressed a kinase-inactive missense mutant of Trl1 (*Wang et al., 2006*), *trl1*-D425N, to assess the contribution of RNA 5'-kinase activity to decay of *HAC1* intermediates. Expression of *trl1*-D425N (lanes 5 and 6) caused similar accumulation of 3'-exon as in the *trl1Δ* deletion, and *xrn1Δ trl1*-D425N cells have levels of 3'-exon similar to *trl1*-D425N alone (compare lanes 6 and 8), indicating that Trl1 5'-kinase activity is required for Xrn1-mediated decay. (**C**) 5'-kinase and ligase domains contribute to the abundance of liberated 3'-exon. We expressed a ligase-inactive missense mutant of Trl1 (*Sawaya et al., 2003*), *trl1*-K114A, to assess the contribution of ligation activity to levels of *HAC1* intermediates. Expression of *trl1*-K114A (lanes 5 and 6) lead to a moderate accumulation of 3'-exon. The double missense mutant, *trl1*-K114A-D425N (lanes 7 and 8), has 3'-exon accumulation much like that of *trl1*-D425N, albeit stronger. (**D**) Dxo1 activity does not affect *HAC1* 3'-exon abundance or cause promiscuous ligation. Northern blot analysis of 3'-exon shows that *dxo1Δ* cells phenocopy wild-type cells, whereas *xrn1Δ* and *dxo1Δ xrn1Δ* cells accumulate similar levels of 3'-exon, indicating that Dxo1 does not contribute substantially to 3'-exon abundance. (**E**) Cells lacking Trl1 and expressing *E. coli* RtcB promiscuously splice *HAC1*ˢ. Northern blot analysis for *HAC1* 5'-exon shows that *trl1Δ* (RtcB) cells promiscuously splice *HAC1*ˢ, similar to *xrn1Δ* cells (compare lanes 5 and 7 to lane 3), and the defects in *HAC1* splicing in *trl1Δ* (RtcB) cells are unaffected by *xrn1Δ* (compare lanes 7 and 8 to 5 and 6). (**F**) RT-PCR assay to measure *HAC1* splicing shows promiscuous splicing of *HAC1*ˢ in *RtcB* and *xrn1Δ* cells. Similar to **E**), here an endpoint RT-PCR assay using primers that flank the intron or splice junction assesses and semi-quantifies *HAC1* splicing. Tunicamycin induces *HAC1*ˢ production in wild-type cells, (lanes 1 and 2) but *HAC1*ˢ is detected in in *trl1Δ* (RtcB) and *xrn1Δ* cells under normal growth conditions (without tunicamycin, lanes 3 and 5). An asterisk marks an unidentified PCR product. (**G**) Model for kinase-mediated decay of cleaved *HAC1* 3'-exon. Ire1 cleavage produces a 3'-exon with a 5'-OH that is phosphorylated by Trl1 5'-kinase. The 5'-phosphorylated product is then adenylylated and ligated to the *HAC1* 5'-exon by Trl1, or degraded by the 5'-phosphate-dependent 5'→3' exonuclease Xrn1. In *xrn1Δ* cells (top), the lack of robust 5'→3' decay favors ligation, leading to promiscuous splicing under normal growth conditions. In *trl1Δ* cells expressing RtcB (bottom), RtcB directly ligates the 5'-OH products of Ire1 cleavage, and the lack of Trl1 5'-kinase activity renders Xrn1 decay irrelevant, causing promiscuous production of *HAC1*ˢ.

DOI: https://doi.org/10.7554/eLife.42262.005

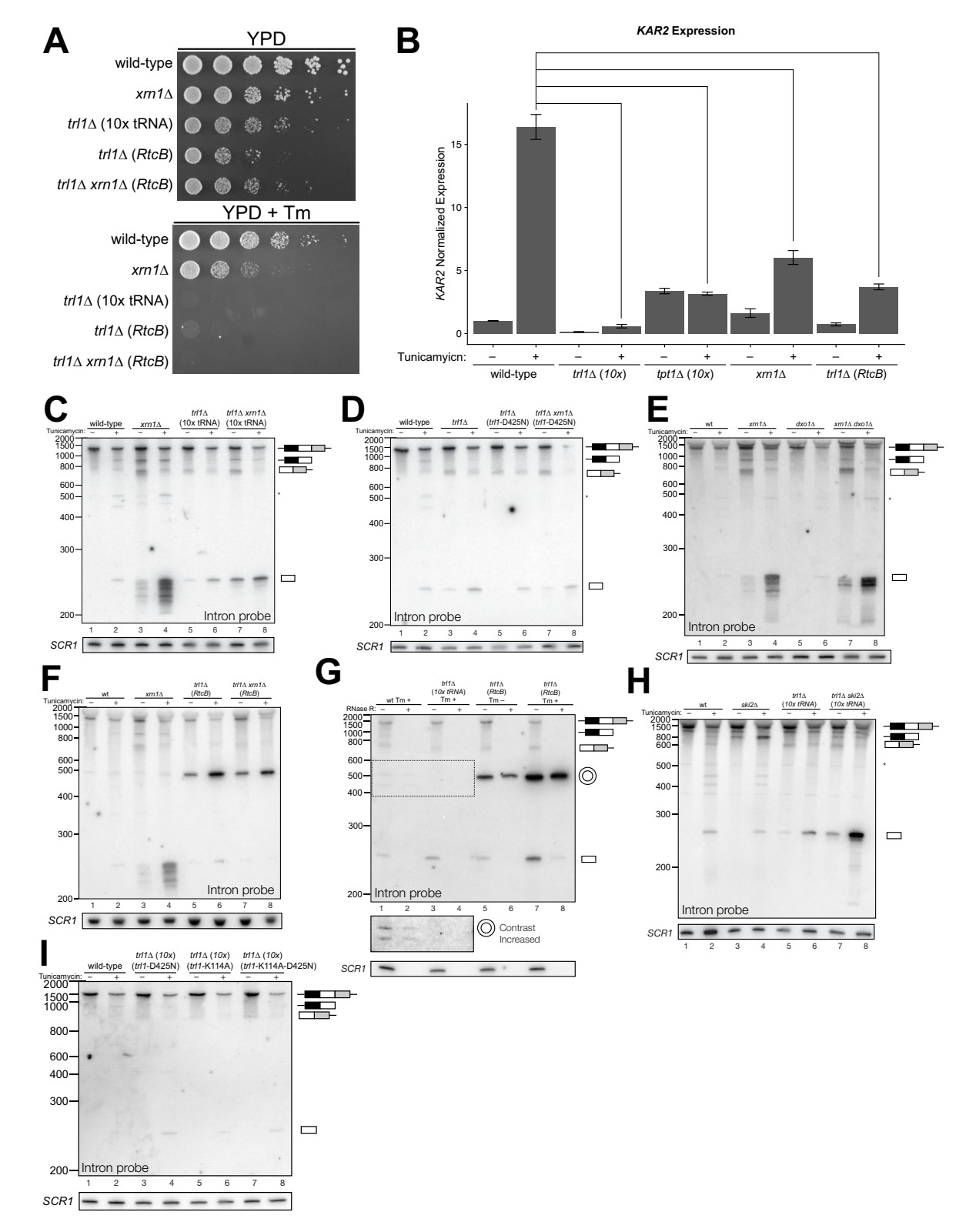

**Figure 3.** Kinase-mediated decay of excised *HAC1* intron is required to activate the unfolded protein response. (**A**) A serial dilution (5-fold) yeast growth assay on rich media (YPD) and tunicamycin-containing media (+Tm) compares the growth of *xrn1Δ*, *trl1Δ* (*10x tRNA*), and *trl1Δ* (*RtcB*) cells to resist protein-folding stress. Growth of wild-type cells is modestly affected by tunicamycin, whereas growth of *xrn1Δ* cells is partially inhibited by tunicamycin. Cells that lack ligase (*trl1Δ*), and cells expressing *E. coli* RtcB RNA ligase in lieu of *TRL1* (*Tanaka et al., 2011*) both fail to grow on media

*Figure 3 continued on next page*

*Figure 3 continued*

containing tunicamycin, and this growth defect is not affected by *xrn1Δ*. (B) Expression of a UPR-responsive gene is compromised in RNA repair and decay mutants. RT-qPCR of mRNA for *KAR2* (BiP), a direct target of Hac1 (**Kohno et al., 1993**), performed on total RNA from the indicated genotypes shows that wild-type cells induce *KAR2* expression by 16-fold upon tunicamycin treatment. (Error bars are 95% confidence interval, *n* = 3; comparison bars represent p<0.01, Student's t-test.) *trl1Δ* cells show an insignificant increase in UPR induction, whereas *tpt1Δ* cells have elevated *KAR2* levels in the absence of tunicamycin, which does not change significantly after tunicamycin treatment. *trl1Δ* (*RtcB*) and *xrn1Δ* cells have a modest increase in expression, but not to the same degree as wild-type (p<0.01). (C) Excised *HAC1* intron is stabilized in *xrn1Δ* and *trl1Δ* cells. Northern blot analysis using a probe to *HAC1* intron reveals that excised intron (252 nt) and partially-degraded intron intermediates accumulate in *xrn1Δ* cells. Ligase-delete cells (*trl1Δ*) also accumulate intron as a uniformly sized 252 nt product. In *xrn1Δ* and *trl1Δ* cells, intron accumulates in the absence tunicamycin. (D) Catalytic activity of Trl1 5′-kinase is required for 5′→3′ decay of excised *HAC1* intron. Northern blot analysis using a probe to *HAC1* intron shows that a missense mutation in the 5′-kinase domain of Trl1 (*trl1*-D425N) phenocopies the *HAC1* accumulation of *trl1Δ* cells (lanes 4 and 6, also C). (E) The distributive 5′→3′ exonuclease Dxo1 can partially degrade *HAC1* intron. Northern blot analysis for *HAC1* intron on total RNA from *dxo1Δ* and *dxo1Δ xrn1Δ* cells shows that Dxo1 can partially degrade *HAC1* intron when it accumulates in *xrn1Δ* cells (compare lanes 4, 6 and 8). (F) A slow-migrating intron species accumulates in *trl1Δ* cells expressing RtcB. Northern blot analysis of RNA from wild-type and *trl1Δ* cells shows that wild-type cells accumulate linear, partially degraded intron (lanes 1–4), whereas *trl1Δ* (*RtcB*), and *trl1Δ xrn1Δ* (*RtcB*) cells accumulate a slower-migrating species (~500 nt; lanes 5–8). (G) Cells expressing RtcB accumulate circular *HAC1* intron. To test whether the slower-migrating band was circularized, total RNA was treated with RNase R and analyzed by northern blot. The slower-migrating species is largely protected from degradation, indicating it is a circle. Linear *HAC1* intron and *SCR1* (bottom) are degraded upon RNase R treatment (lanes 2, 4, 6 and 8). A panel of enhanced contrast shows that the slower migrating species in wild-type cells are circular, excised *HAC1* introns, resistant to RNase R. Circular intron only occurs in samples from cells expressing a ligase. (H) The cytoplasmic exosome degrades *HAC1* intron when 5′→3′ decay is disabled. Northern blot analysis using a *HAC1* intron probe on total RNA from *ski2Δ* (a component of the cytoplasmic exosome) cells showed that *ski2Δ* cells accumulate excised *HAC1* intron (lane 4) at levels similar to wild-type (lane 3). In *trl1Δ ski2Δ* cells, a lack of both kinase-mediated decay (*trl1Δ*) and 3′→5′ decay (*ski2Δ*) causes accumulation of excised intron relative to *trl1Δ* cells (compare lanes 6 and 8). (I) Catalytic activity of Trl1 ligase domain contributes to processing of excised *HAC1* intron. We expressed a ligase-inactive Trl1 allele, *trl1*-K114A, as in **Figure 2C**. Expression of *trl1*-K114A (lanes 5 and 6) lead to a modest accumulation of intron, though not to the same extent as in the kinase-inactivated mutant (*trl1*-D425N) (lanes 3 and 4). The double missense mutant, *trl1*-K114A-D425N (lanes 7 and 8), exhibits intron accumulation similar that of *trl1*-D425N.

DOI: https://doi.org/10.7554/eLife.42262.006

significantly less *KAR2* mRNA upon tunicamycin treatment (**Figure 3B**), consistent with another layer of UPR regulation downstream of HAC1ˢ production. Excised *HAC1* intron accumulates upon tunicamycin treatment in *trl1Δ*, *xrn1Δ*, and *trl1Δ xrn1Δ* cells (**Figure 3C**). The intron decay products that accumulate in these cells are indicative of kinase-mediated decay: in *xrn1Δ* cells that lack Xrn1 but have 5′-kinase activity, intron products appear with a few distinct, smaller products below the full-length 252 nt excised intron (**Figure 3C**, lanes 3 and 4). In contrast, excised intron accumulated as a uniform,~250 nt product in *trl1Δ* cells that lack 5′-kinase activity (**Figure 3C**, lanes 5–8), independent of *XRN1* status. Moreover, production of shorter decay products (like those present in *xrn1Δ* cells) was dependent on Trl1 5′-kinase catalytic activity (**Figure 3D**). Interestingly, expression of the adenylyl-transferase-dead/ligase-dead allele, *trl1*-K114A, also led to accumulation of some free *HAC1* intron (**Figure 3I**), potentially indicating a role for ligation in the processing of liberated *HAC1* intron. Together, these data show that excised *HAC1* intron is a substrate for kinase-mediated decay with strict dependence on a 5′-phosphorylation step to promote 5′→3′ decay.

A single deletion of Dxo1 had no effect on *HAC1* intron degradation (**Figure 3E**); however, the sizes of smaller decay products in *xrn1Δ dxo1Δ* cells were subtly different than in *xrn1Δ* cells (**Figure 3E**), indicating that Dxo1 and other exonucleases partially degrade excised *HAC1* intron, but only when it accumulates in *xrn1Δ* cells. Consistent with this notion, we found that the cytoplasmic exosome also contributes to *HAC1* intron turnover (**Figure 3H**), but this mode of decay is unlikely to regulate the UPR as the growth of *ski2Δ* cells is unaffected by tunicamycin (**Figure 1B**).

In *trl1Δ* (*RtcB*) cells, excised *HAC1* intron accumulates as a circle, evinced by its altered mobility and resistance to Xrn1-mediated decay in vivo (**Figure 3F**), and its resistance to RNase R degradation in vitro (**Figure 3G**). Circularization of the *HAC1* intron by RtcB in *trl1Δ* cells is facilitated by 5′-OH and 2′,3′-cyclic PO4 termini created by Ire1 cleavage and the absence of Trl1 end modification activities that could otherwise produce termini incompatible with RtcB ligation (5′-PO4 or adenylylate; and 2′-PO4/3′-OH). It is noteworthy that circularized intron accumulates to high levels in the absence of tunicamycin (**Figure 3F and G**), indicating that Ire1 catalyzes a low level of intron excision (and 3′-exon excision (**Figure 1C**)) from HAC1ᵘ during normal growth, leading to the accumulation of stable, circularized introns in the presence of RtcB.

The *HAC1* intron and 5′-UTR form an extensive base-pairing interaction that inhibits ribosome initiation (*Chapman and Walter, 1997*; *Di Santo et al., 2016*). Thus, together these data evoke a model in which kinase-mediated decay of the excised *HAC1* intron is required for *HAC1*ˢ translation, and a failure to degrade *HAC1* intron—even when *HAC1*ˢ is produced—prevents *HAC1*ˢ translation and subsequent expression of stress-responsive genes. Despite their ability to make *HAC1*ˢ, *xrn1Δ* and *trl1Δ* (*RtcB*) cells have growth defects on media containing tunicamycin (*Figure 3A*) and the relative severity of their defects parallels the accumulation of excised intron in these cells. Cells lacking Xrn1 have a modest growth defect and accumulate linear intron, which can be degraded by other exonucleases (*Figure 3*). In contrast, *trl1Δ* (*RtcB*) cells have a severe growth defect and accumulate high levels of a stable, circularized intron that is immune to exonucleolytic decay (*Figure 3F and G*). We believe these findings resolve the mystery of the previously identified 'second function' of Trl1 required for UPR activation (*Mori et al., 2010*), namely that—in addition to its ligase activity—Trl1 initiates kinase-mediated decay of the excised *HAC1* intron, relieving its repressive effect on *HAC1*ˢ and activating translation.

## Incompletely processed *HAC1*ˢ mRNA is endonucleolytically cleaved and degraded

In addition to the canonical 5′-exon product of 5′-splice site cleavage, a second product uniquely accumulates in *tpt1Δ* and *tpt1Δ ski2Δ* cells that is ~50 nt shorter than full-length 5′-exon (*Figure 4A*). Expression of a catalytically inactive form of Tpt1 (Tpt1-R138A, (*Sawaya et al., 2005*)) in *tpt1Δ* cells failed to rescue this defect (*Figure 4B*), affirming that the catalytic activity of Tpt1 is required to prevent accumulation of the shorter 5′-exon fragment. A corresponding elongated 3′-exon fragment accumulates in *tpt1Δ*, and more intensely in *tpt1Δ xrn1Δ* cells, indicating it is degraded by Xrn1 (*Figure 4C–E*). The elongated 3′-exon is specifically detected using a northern probe with a sequence complementary to the distal 3′-end of the 5′-exon (*Figure 4C,D*), indicating that a portion of the 5′-exon is responsible for the increased size of this decay intermediate. Moreover, a fragment of similar size hybridizes to a probe for the 3′-exon, suggesting that the sequence derived from 5′-exon is linked to the 3′-exon (*Figure 4E*); together these data indicate that *HAC1*ˢ is cleaved upstream of the 2′-phosphorylated ligation junction in *tpt1Δ* cells, and these products are degraded by both Xrn1 and the cytoplasmic exosome.

To determine whether *HAC1*ˢ sequence is sufficient for cleavage, we expressed plasmid-encoded *HAC1*ᵘ and *HAC1*ˢ in *hac1Δ* cells to identify *HAC1* splicing intermediates. We performed the analysis on *trl1Δ* cells, reasoning that if *HAC1*ˢ sequence were sufficient to cause cleavage, we would expect an accumulation of 3′-exon in cells unable to carry out ligation or degrade the products by kinase-mediated decay. Tunicamycin treatment of *trl1Δ hac1Δ* cells expressing *HAC1*ᵘ caused Ire1-mediated cleavage and accumulation of cleaved 3′-exon (*Figure 4F*). However, we found that full length *HAC1*ˢ was the only product produced in the *trl1Δ hac1Δ* cells in the presence and absence of tunicamycin (*Figure 4F*). Furthermore, expression of *HAC1*ˢ in a *tpt1Δ* background failed to produce additional fragments (*Figure 4G*), whereas expression of *HAC1*ᵘ is sufficient in the *tpt1Δ* background to produce free *HAC1* 3′-exon, consistent with its secondary cleavage. Together, these results (*Figure 4F and G*) indicate that the *HAC1*ˢ transcript is not sufficient to recapitulate the secondary cleavage, and that *HAC1*ᵘ must be cleaved and ligated to produce the 2′-phosphorylated *HAC1*ˢ secondary cleavage substrate.

To further characterize this secondary cleavage product, we determined the 5′ end of the elongated 3′-exon. We analyzed the 5′-ends of cleaved 3′-exon by primer extension and found that cleaved 3′-exon was barely detectable in wild-type cells upon tunicamycin addition, whereas 3′-exon accumulated in *trl1Δ* cells due to a lack of kinase-mediated decay (*Figure 5A*). In *tpt1Δ* cells treated with tunicamycin, two products accumulated upon tunicamycin addition: a product consistent with canonical length 3′-exon and a small amount of elongated 3′-exon (*Figure 5A*). The elongated product accumulated in *tpt1Δ xrn1Δ* cells, again indicating it is degraded by Xrn1 (*Figure 5A*). To test this prediction (summarized in *Figure 5B*), we measured the susceptibility of 3′-exon fragments to treatment in vitro with recombinant Xrn1 (rXrn1/TEX). As expected, fragments from *xrn1Δ* cells were degraded by rXrn1 (*Figure 5C*), establishing that they have 5′-PO$_4$ termini, whereas 3′-exon fragments from *trl1Δ* cells were resistant to rXrn1 degradation (*Figure 5C*), indicating that they have 5′-OH termini.

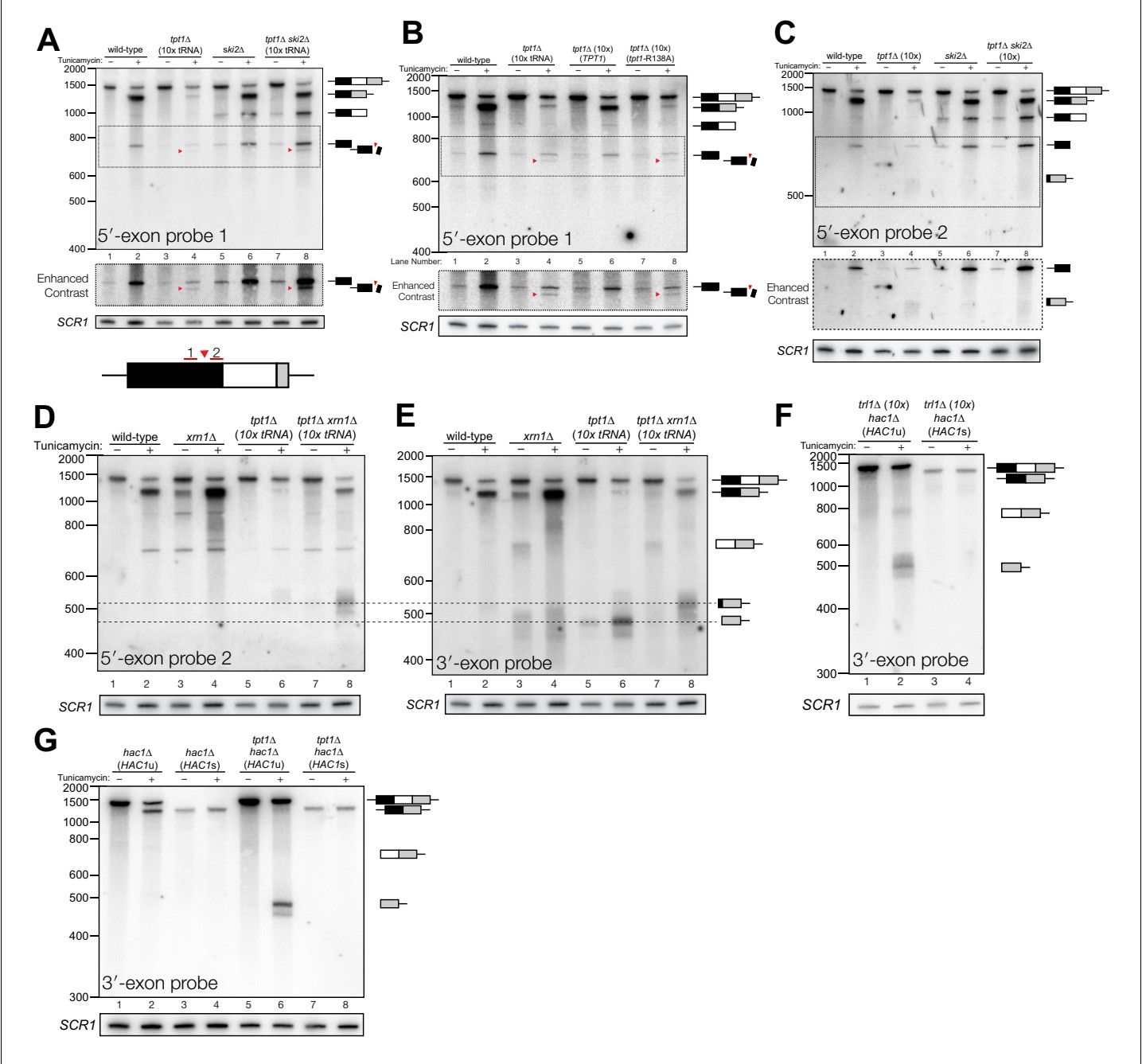

**Figure 4.** Incompletely processed *HAC1s* mRNA is cleaved and degraded. (**A**) A shortened form of 5′-exon accumulates in *tpt1Δ* cells and is degraded by the cytoplasmic exosome. Northern blot analysis of *tpt1Δ* and *tpt1Δ ski2Δ* using a probe to 5′-exon identified a shortened form of liberated 5′-exon, about 50 nt smaller than full-length (red arrowheads). A region of enhanced contrast shows the specific accumulation of this product in *tpt1Δ* and *tpt1Δ ski2Δ* cells. The bottom diagram depicts the relative positions of probes 1 and 2 on the *HAC1* 5′-exon; a red arrowhead marks the putative site of cleavage. (**B**) Production of shortened 5′-exon requires the catalytic activity of Tpt1. Northern blot analysis of *tpt1Δ* cells expressing either a plasmid-encoded wild-type copy of *TPT1* or a catalytically-inactive missense mutant (*tpt1*-R138A) (**Sawaya et al., 2005**) using a probe for 5′-exon. Production of the cleaved 5′-exon seen in *tpt1Δ* cells (lane 4, red arrow marks the band) is rescued by plasmid-mediated expression of wild-type *TPT1* (lane 6) but not by Tpt1-R138A (lane 8, red arrow marks the band), confirming that Tpt1 catalytic activity is required to prevent *HAC1s* secondary cleavage. (**C**) The shortened 5′-exon is missing a portion of its 3′-end. A northern blot was probed with 5′-exon probe 2, which hybridizes to the 3′-most 30 nt of *HAC1* 5′-exon (see diagram in A). Compared to the blot in (**A**), the shortened 5′-exon species is absent (lanes 4 and 8) and instead the probe detects smaller bands consistent with the length of the elongated 3′-exon (see region of enhanced contrast). (**D**) A lengthened form of 3′-exon accumulates in *tpt1Δ* cells. Total RNA from *tpt1Δ* and *xrn1Δ* cells was analyzed by northern blot with 5′-exon probe 2, a probe that anneals to the 3′-most 30 nt of *HAC1* 5′-exon. (**D**) and E) share the same band interpretation key, so dashed lines have been drawn across to illustrate the different positions of typical 3′-exon
*Figure 4 continued on next page*

*Figure 4 continued*

(474 nt) and elongated 3′-exon (524 nt). (E) The elongated 3′-exon from *tpt1Δ* cells co-migrates and co-hybridizes with 5′-exon probe 2 (see D). Stripping the blot from D) and re-hybridizing it with *HAC1* 3′-exon probe detects the same, elongated band(s) as in the 5′-exon probe two blot (lane 8), as well as *HAC1* 3′-exon bands of typical length. (F) Expression of 'pre-spliced' *HAC1*ˢ is not sufficient to promote cleavage. Total RNA from *trl1 hac1Δ* cells expressing *HAC1*ᵘ and *HAC1*ˢ from a plasmid was analyzed by northern blot with a 3′-exon probe. Full length *HAC1*ᵘ expressed from this construct is cleaved upon tunicamycin addition (lane 2). Expression of 'pre-spliced' *HAC1*ˢ yields a single product with no decay intermediates (lanes 3 and 4), indicating that 'pre-spliced' *HAC1*ˢ—which was not produced by ligation and therefore lacks a 2′-phosphate—is not sufficient to recapitulate the cleavage found in *tpt1Δ* cells. (G) Expression of 'pre-spliced' *HAC1*ˢ is not sufficient to promote secondary cleavage. Total RNA from *hac1Δ* and *tpt1Δ hac1Δ* cells expressing *HAC1*ᵘ and 'pre-spliced' *HAC1*ˢ from a plasmid was analyzed by northern blot with a 3′-exon probe. In *hac1Δ* cells, full length *HAC1*ᵘ expressed from this construct is cleaved and ligated upon tunicamycin addition (lane 2). Expression of 'pre-spliced' *HAC1*ˢ yields a single product with no processing intermediates (lanes 3 and 4), indicating that 'pre-spliced' *HAC1*ˢ is not further processed. The *HAC1*ᵘ construct expressed in cells of genotype *tpt1Δ hac1Δ* (*10x tRNA*) gets cleaved upon tunicamycin treatment (lane 6), yielding a secondary cleavage fragment. Expression of pre-spliced *HAC1*ˢ in *tpt1Δ hac1Δ* cells does not produce any additional processing fragments, indicating that 2′-phosphorylated products from *HAC1*ᵘ are required for secondary cleavage in *tpt1Δ* cells.

DOI: https://doi.org/10.7554/eLife.42262.007

The instructive findings came from examining 3′-exon accumulation in *tpt1Δ* cells. The 3′-exon fragment of canonical length that accumulates in *tpt1Δ* cells was resistant to rXrn1 treatment (*Figure 5C*), while the elongated fragment of 3′-exon is susceptible to rXrn1 treatment (*Figure 5C*, lanes 15 and 16). These observations indicate that 3′ product of secondary cleavage of *HAC1*ˢ is a substrate of Xrn1 in vivo and in vitro, raising the possibility that it may also be a substrate of kinase-mediated decay, depending on the chemistry of the endoribonuclease that generates the secondary cleavage. We propose these products are created via two steps: (*i*) *HAC1*ˢ is cleaved ~50 nt upstream of the 2′-phosphorylated ligation junction; (*ii*) Xrn1 partially degrades the intermediate fragment to the site of 2′-phosphorylation, which inhibits further degradation (*Figure 5D*). Under this model, the 3′-exon fragment that accumulates in *tpt1Δ* cells has both 5′-PO₄ and 2′-PO₄ moieties at its first position, which inhibits Xrn1-mediated decay in vivo and in vitro (*Figures 1C*, *4D*, *5A and C*).

The accumulation of *HAC1* decay intermediates in *tpt1Δ* cells over time further supports a model of *HAC1*ˢ cleavage by Ire1. In *tpt1Δ* cells, the accumulation of secondary cleavage product coincides with the increase in production of *HAC1*ˢ at 20 min (*Figure 6*). In *tpt1Δ* cells, cleaved *HAC1* 3′-exon is present at low levels at steady state and accumulates over the course of two hours upon tunicamycin treatment (*Figure 6A*). *HAC1*ˢ is also generated in *tpt1Δ* cells, but at significantly reduced levels compared to wild-type (*Figure 6A*). It is also notable that *tpt1Δ xrn1Δ* cells and *tpt1Δ ski2Δ* cells accumulate more spliced *HAC1*ˢ than *tpt1Δ* cells (*Figure 6E*), suggesting that some *HAC1*ˢ molecules or splicing intermediates in *tpt1Δ* cells are degraded, possibly because they contain 2′-PO₄ moieties. At all time points, *tpt1Δ* cells contain more free 3′-exon than *HAC1*ˢ, a ratio opposite to wild-type cells (*Figure 6A*), indicating that that 3′-exon cleaved from *HAC1*ᵘ accumulates as a result of partial decay of 5′- and 2′-phosphorylated *HAC1*ˢ. The augmented accumulation of 3′-exon in *tpt1Δ* cells is also observed in the primer extension analysis (*Figure 5A*). The in vitro ability of Xrn1 to only partially degrade elongated *HAC1* 3′-exon from *tpt1Δ xrn1Δ* cells, and inability to degrade canonical 3′-exon from *tpt1Δ* cells, indicates that the accumulation of free *HAC1* 3′-exon is likely caused primarily by blocked 5′→3′ degradation.

## Discussion

Many different regulatory events impinge on *HAC1* mRNA to control its localization and processing. It has been assumed that cleavage of *HAC1*ᵘ by Ire1 is the rate-limiting step for UPR activation. Counter to this view, we found that decay of *HAC1* splicing intermediates is required for both UPR activation and suppression. We found several examples wherein 'kinase-mediated decay' (KMD) degrades *HAC1* splicing intermediates containing 5′-OH termini by sequential 5′-phosphorylation and 5′-phosphate-dependent 5′→3′ exonucleolytic degradation (*Figure 7*).

We propose that after 3′-splice site cleavage by Ire1, the Trl1 5′-kinase domain associates with and phosphorylates the 5′-OH of the 3′-exon product (*Figure 7*). Dissociation of the Trl1 kinase active site from the 5′-PO₄ product then enables a competition between reassociation of Trl1 (now

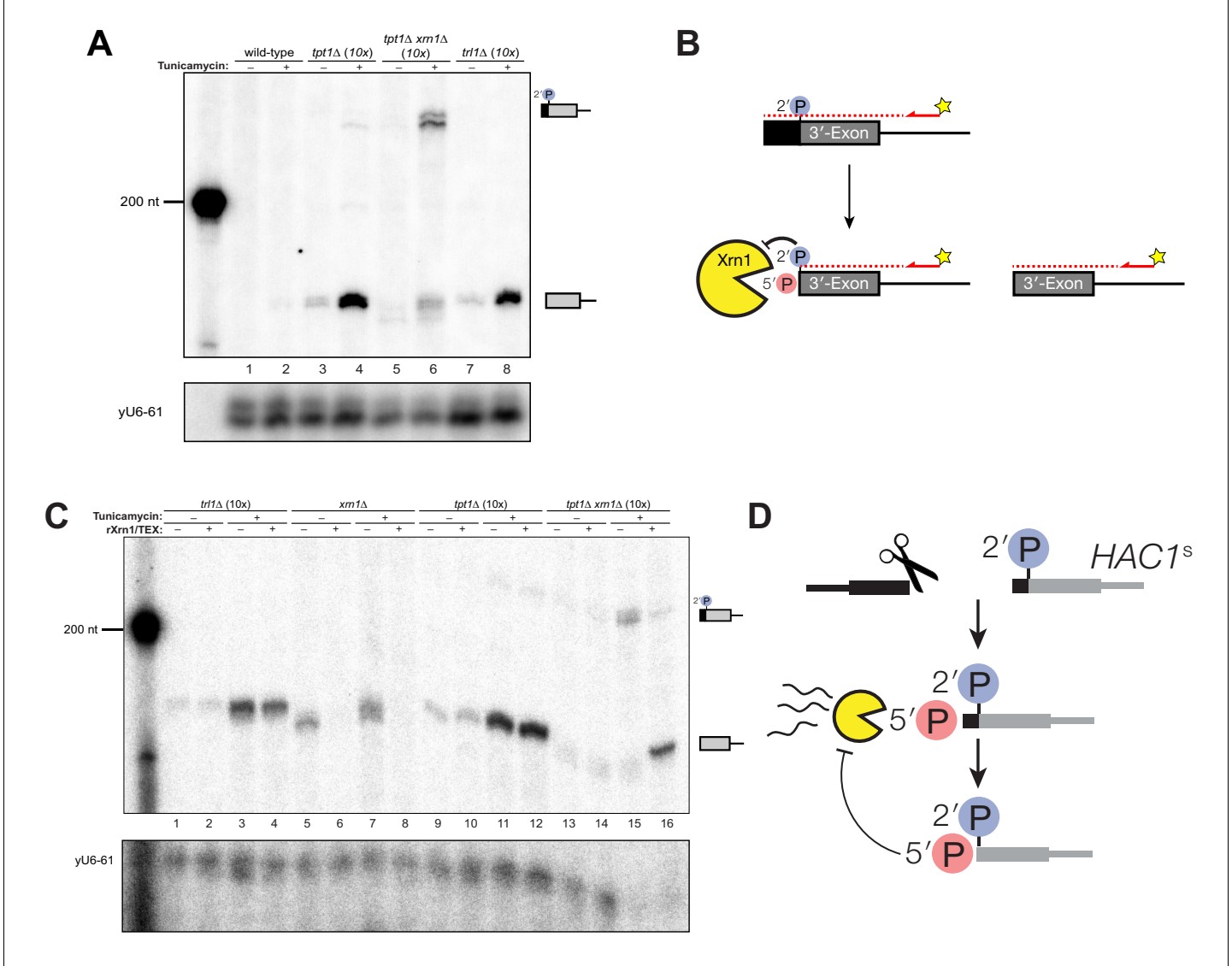

**Figure 5.** A 5'- and 2'-phosphorylated *HAC1* decay intermediate inhibits Xrn1. (**A**) Primer extension analysis of *HAC1* 3'-exon cleavage products. Primer extension using a probe for 3'-exon was performed on total RNA from wild-type, *tpt1Δ* (*10x tRNA*), *tpt1Δ xrn1Δ* (*10x tRNA*), and *trl1Δ* (*10x tRNA*) cells treated with or without tunicamycin. A loading control for U6 snRNA is depicted in the bottom panel. The extension product of cleaved 3'-splice site is 174 nt, found in wild-type cells treated with tunicamycin (lane 2) and is present in untreated *trl1Δ* cells but increases upon tunicamycin treatment (lanes 7 and 8). An extension product from *tpt1Δ* cells accumulates upon tunicamycin treatment (lanes 3 and 4) and co-migrates with the product from wild-type (lane 2) and *trl1Δ* (lane 8) cells. In addition, a faint elongated product at ~225 nt is present in *tpt1Δ* cells (lane 4). This elongated product accumulates to higher levels in *tpt1Δ xrn1Δ* cells treated with tunicamycin (lane 6). (**B**) Model of *HAC1* 3'-exon primer extension product lengths. A 5'-radiolabeled (yellow star) primer (*HAC1* 3'-exon probe, see *Table 2*) anneals to *HAC1* 3'-exon mRNA and primes cDNA synthesis by a reverse transcriptase. The reverse transcriptase stops synthesizing cDNA when it runs out of RNA template at the 5'-terminus of the 3'-exon. The model shows three situations as observed in (**A**): 1) canonical 3'-exon cleavage fragments (lower right), as observed in *trl1Δ* cells; 2) extended *HAC1* 3'-exon (above reaction arrow), as caused by secondary cleavage of *HAC1*s, observed in *tpt1Δ xrn1Δ* (*10x*) cells; and 3) extended *HAC1* 3'-exon on which Xrn1 (cellular or in vitro) initiated decay, but failed to proceed past the 2'-P (beneath arrow). (**C**) Susceptibility of 3'-exon cleavage products to in vitro Xrn1 degradation. Total RNA from (**A**) was treated with recombinant Xrn1 (rXrn1) and analyzed by primer extension for the 3'-exon. The loading control performed on U6 snRNA is depicted in the bottom panel. The 3'-exon from *trl1Δ* cells accumulates upon tunicamycin treatment (compare lanes 1 and 2 to 3 and 4) but the 3'-exon is resistant to rXrn1 (lanes 3 and 4) because it lacks a 5'-phosphate due to lack of Trl1 5'-kinase activity in these cells. In contrast, the 3'-exon products from *xrn1Δ* cells, which have Trl1 5'-kinase and thus 5'-phosphates, are degraded by rXrn1 (lanes 6 and 8). The 3'-exon product in *tpt1Δ* cells is resistant to rXrn1 treatment (compare lanes 9 to 10, and 11 to 12), despite the fact these cells have both Trl1 and Xrn1. The elongated 3'-exon product that accumulates in *tpt1Δ xrn1Δ* cells is partially degraded by rXrn1 (compare lane 15 to 16) and the decay intermediate co-migrates with cleaved 3'-exon. (**D**) Model depicting cleavage and decay of 2'-phosphorylated *HAC1*s. *HAC1*s is cleaved (likely by Ire1)~50 nt upstream of the 2'-phosphorylated ligation junction, creating a 3'-product with ~50 nt of sequence of the 5'-exon (black) and an internal 2'-phosphate. Xrn1

*Figure 5 continued on next page*

*Figure 5 continued*

initiates decay at the 5′-terminus, degrading the 5′-exon portion (black) up to the site of 2′-phosphorylation. The product of partial decay contains a 5′- and 2′-phosphate at its first position, which inhibits further degradation by Xrn1.

DOI: https://doi.org/10.7554/eLife.42262.008

its adenylyltransferase/ligase domain) to catalyze ligation—or Xrn1 to catalyze degradation. In some circumstances, this balance is tipped to favor ligation even in the absence of overt UPR stimulation: a lack of decay in *xrn1Δ* cells favors ligation, whereas the lack of 5′-kinase activity in *trl1Δ* (*RtcB*) cells renders Xrn1 decay irrelevant (*Figure 2E and F*). Xrn1 is abundant in budding yeast (*Ghaemmaghami et al., 2003*), which may efficiently suppress the UPR under normal conditions by degrading spuriously cleaved *HAC1* 3′-exon intermediates. Regulation of the ligation step of *HAC1*ˢ splicing makes intuitive sense because it is the last opportunity to act during splicing; once ligated, *HAC1*ˢ is again protected by a 7-methylguanosine cap and a poly-(A) tail. Spurious *HAC1* splicing was previously reported in *trl1Δ* cells expressing the tRNA ligase from *Arabidopsis thaliana* (*Mori et al., 2010*). It is noteworthy that there are mechanistic differences between the tRNA ligases of *A. thaliana* and another yeast (*K. lactis*) (*Remus and Shuman, 2014*), suggesting that these differences could impact the balance between kinase-mediated decay and ligation in budding yeast expressing plant RNA ligase.

We also found that excised *HAC1* intron is a substrate of kinase-mediated decay (*Figure 3*). Indeed, we believe that phosphorylation of *HAC1* intron by Trl1 to promote kinase-mediated decay is the previously proposed 'second role' of Trl1 ligase in activating *HAC1* translation independent of ligation (*Mori et al., 2010*). In this previous study, excised and circularized *HAC1* intron was found to remain associated with *HAC1*ˢ, inhibiting translation. Kinase-mediated decay of excised intron therefore likely relieves the long-range base-pairing interaction that prevents *HAC1*ˢ translation (*Chapman and Walter, 1997*; *Di Santo et al., 2016*; *Rüegsegger et al., 2001*), explaining how *HAC1*ˢ can accumulate without concomitant UPR activation. This second layer of control over *HAC1*ˢ translation by KMD adds another failsafe mechanism to prevent its translation and unintentional UPR activation.

Previous examples of kinase-mediated decay of bacterial mRNAs (*Durand et al., 2012*), eukaryal tRNA introns (*Wu and Hopper, 2014*), ribosomal RNA processing intermediates (*Gasse et al., 2015*), and no-go mRNA decay cleavage products (*Navickas et al., 2018*) suggest that this mode of decay may be widespread. Coupling of RNA 5′-kinase and 5′→3′ exonucleolytic decay activities in the context of kinase-mediated decay may regulate the UPR in other organisms. Splicing of Xbp-1 mRNA in metazoans (the functional homolog of *HAC1*) is catalyzed by Ire1-mediated removal of an

**Table 2.** Oligonucleotide sequences.

| Oligonucleotide name | Oligonucleotide sequence (5′→3′) |
|---|---|
| *HAC1*-F RT-PCR | ACCTGCCGTAGACAACAACAAT |
| *HAC1*-R RT-PCR | AAAACCCACCAACAGCGATAAT |
| *KAR2* qPCR F | AAGACAAGCCACCAAGGATG |
| *KAR2* qPCR R | AGTGGCTTGGACTTCGAAAA |
| *PGK1* qPCR F | TCTTAGGTGGTGCCAAAGGTT |
| *PGK1* qPCR R | GCCTTGTCGAAGATGGAGTC |
| *HAC1* 5′-exon probe 1 | AAGTCTCTTGGTCCGACGCGGAATCGCGCA |
| *HAC1* 5′-exon probe 2 | CTGGATTACGCCAATTGTCAAGATCAATTG |
| *HAC1* intron probe 1 | AACCGGCTCCTCCCCCATCAGAGAACCACGA |
| *HAC1* intron probe 2 | GGACAGTACAAGCAAGCCGTCCATTTCTTAGT |
| *HAC1* 3′-exon probe (primer extension and northern) | ACCGGAGACAGAACAGTAGAAACCACTAAGCG |
| *KAR2* probe | ACCGTAGGCAATGGCGGCTGCGGTTGGTTC |
| *SCR1* probe (oRP100) | GTCTAGCCGCGAGGAAGG |

DOI: https://doi.org/10.7554/eLife.42262.011

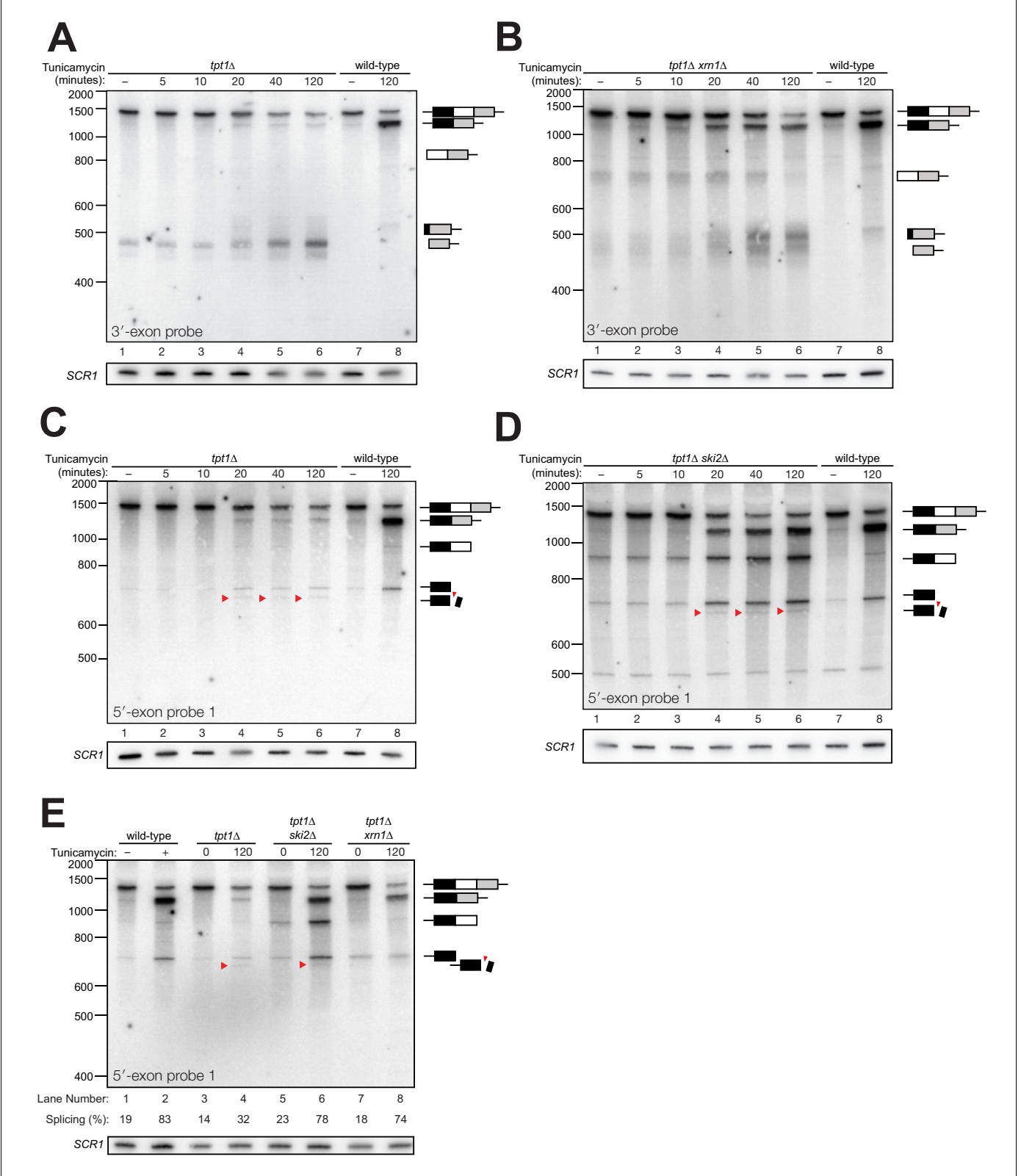

**Figure 6.** Kinetic analysis of *HAC1* mRNA processing in cells lacking Tpt1. (**A**) Northern blot analysis of a time course of tunicamycin treatment (0 to 120 min) in *tpt1Δ* (*10x tRNA*) cells showing the dynamics of *HAC1*$^u$ splicing using a probe for 3′-exon. The *SCR1* loading control is shown below the panel. RNA from wild-type cells treated for 0 and 120 min tunicamycin was loaded in lanes 7 and 8. *HAC1*$^s$ accumulates slowly over 120 min and its abundance at the end of the time-course is 35-fold lower than wild-type (lanes 6 and 8). A 3′-exon cleavage product (~475 nt) is present at time 0 and

*Figure 6 continued on next page*

*Figure 6 continued*

accumulates over time; its increase is coincident with the appearance of *HAC1ˢ* (20 min time point) and its abundance at 120 min exceeds the level of *HAC1ˢ*. A 3′-exon cleavage product in wild-type cells (~500 nt) is apparent at 120 min (lane 8). (B) Northern blot analysis of *HAC1* splicing in *tpt1Δ xrn1Δ* (*10x tRNA*) cells using a probe for 3′-exon. Conditions are the same as A). *HAC1ˢ* increases at 20 min and its final level (lane 6) is higher than *HAC1ˢ* in *tpt1Δ* cells (A), lane 6), approaching wild-type levels (lane 8). Levels of the product of 5′-splice site cleavage (intron/3′-exon) accumulate over 120 min. The 3′-exon product (~475 nt) and cleaved 3′-exon (~500 nt) begin accumulating at 20 min, coincident with increased levels of *HAC1ˢ*. A 3′-exon cleavage product in wild-type cells (~500 nt) is apparent at 120 min (lane 8). (C) Northern blot analysis of *HAC1* splicing in *tpt1Δ* (*10x tRNA*) cells using a probe for 5′-exon. Conditions are the same as A). *HAC1ˢ* increases at 20 min up to final level. Canonical 5′-exon (728 nt) and secondarily-cleaved 5′-exon (~675 nt; red arrowheads) begin accumulating at 20 min, coincident with increased levels of *HAC1ˢ*. (D) Northern blot analysis of *HAC1* splicing in *tpt1Δ ski2Δ* (*10x tRNA*) cells using a probe for 5′-exon. Deletion of *SKI2* in the context of *tpt1Δ* increases the abundance of nearly all intermediates relative to *tpt1Δ* alone (A). Shortened *HAC1* 5′-exon is marked with red arrowheads. The product of 3′-splice site cleavage (5′-exon/intron) is present at time 0 and accumulates over 120 min, whereas *HAC1ˢ* accumulates beginning at 20 min. (E) Northern blot analysis of *HAC1* splicing using a probe for 5′-exon to compare the beginning and end time points of time course experiments performed on *tpt1Δ*, *tpt1Δ xrn1Δ*, and *tpt1Δ ski2Δ*. Results from densitometry of the spliced and unspliced bands are written beneath the blot. Less *HAC1* splicing takes place in *tpt1Δ* cells compared to wild type, and disabling decay factors *ski2Δ* and *xrn1Δ* in the *tpt1Δ* mutant lead to increased *HAC1* splicing, compared to *tpt1Δ* alone.

DOI: https://doi.org/10.7554/eLife.42262.009

intron and ligation by RtcB RNA ligase (*Kosmaczewski et al., 2014*; *Lu et al., 2014*). Fundamental differences in the chemistry of RNA ligation between fungal and metazoan ligases suggest that Xbp-1 may be subject to a biochemically distinct mode of regulation. Because RtcB depends on 5′-OH and 3′-PO₄ termini for catalysis (*Chakravarty et al., 2012*), activities that remodel 5′-OH RNA termini could divert Ire1-generated 5′-OH splicing intermediates from productive ligation. To that point, cyclic nucleotide phosphodiesterase (CNP) and RtcA (a 2′,3′-RNA cyclase) were shown to 'tune' the UPR in metazoans by competing with RtcB/HSCP117 for ligation substrates (*Unlu et al., 2018*). Specifically, CNP hydrolyzes the 2′,3′-cyclic phosphate compatible RtcB to a 2′-PO₄ incompatible with RtcB, decreasing ligation of Xbp1. Conversely, RtcA, which converts 2′-PO₄ RNA to 2′,3′-cyclic phosphate, makes the terminus compatible with RtcB, thus enhancing Xbp1 splicing.

Additionally, an RNA 5′-kinase (*e.g.*, Clp1) may phosphorylate the 3′-exon product of Xbp-1 cleavage, simultaneously inhibiting ligation by RtcB and promoting its degradation by a 5′-phosphate-dependent exoribonuclease to limit UPR activation. In this vein, it is noteworthy that kinase-inactivating mutations in Clp1 cause neurodevelopmental defects and neuronal dysfunction in humans, mice, and zebrafish (*Hanada et al., 2013*; *Karaca et al., 2014*; *Schaffer et al., 2014*), possibly due to chronic UPR activation in neural tissues (*Clayton and Popko, 2016*). Xrn1 degrades mRNA fragments generated during metazoan Regulated Ire1-dependent Degradation (RIDD) (*Hollien and Weissman, 2006*); however, it is not known how these 5′-OH cleavage products of Ire1 are phosphorylated for Xrn1-mediated decay. The RNA 5′-kinase Clp1 (*Weitzer and Martinez, 2007*) and polynucleotide kinase Nol9 (*Heindl and Martinez, 2010*) are candidates for this activity, though neither are known to phosphorylate mRNA decay intermediates.

While 5′→3′ decay plays a major role in UPR regulation, we found little evidence for UPR regulation by 3′→5′ decay activity. As shown previously (*Schwer et al., 2004*), the exposed 3′-end of cleaved *HAC1* 5′-exon is a substrate for 3′→5′ decay (*Figure 4*). But while excised *HAC1* intron is stabilized in *ski2Δ* cells lacking cytosolic 3′→5′ decay (*Figure 3H*), their growth is unaffected by tunicamycin (*Figure 1B*), indicating that 3′→5′ decay of the excised intron does not contribute substantially to intron-mediated *HAC1ˢ* repression.

We also found evidence that incompletely processed *HAC1ˢ* mRNA is cleaved, which is ligated but contains an internal 2′-PO₄ moiety. Cleavage of *HAC1ˢ* leads to 5′ and 3′ fragments that are degraded; however, the 3′-fragment is only partially degraded by kinase-mediated decay, producing a 5′- and 2′-phosphorylated molecule that cannot be degraded by Xrn1. Consistent with these findings, a recent study also showed that an RNA with an internal 2′-phosphate group is protected from 3′→5′ decay by *E. coli* PNPase in vitro (*Munir et al., 2018*); those and our results together indicate that site-specific installation of a 2′-PO₄ is an effective strategy to protect an RNA from complete exonucleolytic degradation in vivo or in vitro.

Decay intermediates produced from incompletely processed, 2′-phosphorylated *HAC1ˢ* have not been previously observed and suggest a plausible regulatory role for Tpt1 in regulating *HAC1ˢ* fate. We have yet to determine the impact of 2′-phosphorylation on *HAC1ˢ* translation, but given that

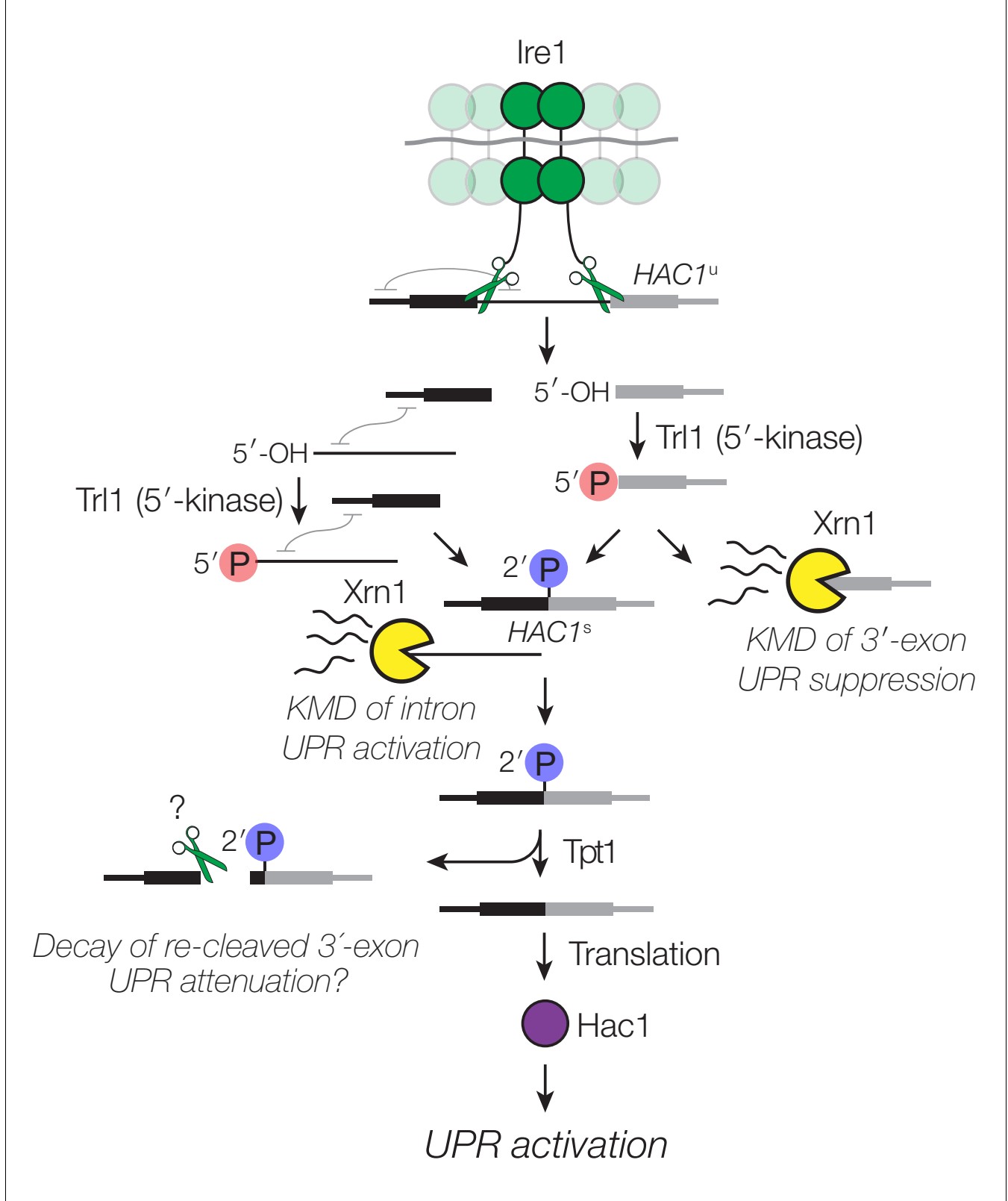

**Figure 7.** Decay of *HAC1* splicing intermediates regulates UPR activation, suppression, and attenuation. Activation of Ire1 by ER stress activates its endoribonuclease activity, leading to cleavage of the 5′- and 3′-splice sites of *HAC1*$^u$. The cleaved 3′-exon (grey) is phosphorylated by the 5′-kinase activity of Trl1, permitting either its ligation or decay by Xrn1. The 5′-exon (black) and intron (thin line) form an extensive base-pairing interaction (grey squiggle). Kinase-mediated decay of the intron is required to activate the translation of *HAC1*$^s$. Ligated but 2′-phosphorylated *HAC1*$^s$ may also be
*Figure 7 continued on next page*

*Figure 7 continued*
cleaved by Ire1 upstream of the ligation junction, and the cleavage products are degraded by Xrn1 and the cytoplasmic exosome. In cells lacking Tpt1, a unique KMD intermediate accumulates with 5′- and 2′-phosphates, which inhibit Xrn1-mediated degradation.
DOI: https://doi.org/10.7554/eLife.42262.010

*tpt1Δ* cells grow on tunicamycin (*Figure 1B* and *Cherry et al., 2018*) and activate *KAR2* gene expression (*Figure 3B*), it is likely that some Hac1 protein is produced from 2′-$PO_4$ *HAC1*$^s$ mRNA. It also remains to be determined how and why incompletely processed *HAC1*$^s$ is cleaved. Insofar as the *HAC1*$^s$ cleavage substrate is initially produced by tunicamycin-dependent Ire1 cleavage and ligation, we conjecture that Ire1 incises ligated, 2′-phosphorylated *HAC1*$^s$ upstream of the original ligation junction, yielding smaller 5′-exon and larger 3′-exon fragments. Formally, we cannot currently rule out the possibility that another endonuclease catalyzes secondary *HAC1*$^s$ cleavage; however, Ire1 is the only endoribonuclease known to site-specifically incise *HAC1* mRNA. We note that *ire1Δ* mutants are unable to initiate processing of *HAC1*$^u$ and therefore do not make *HAC1*$^s$ in the first place (*Sidrauski and Walter, 1997*), precluding direct analysis of *HAC1*$^s$ cleavage in *ire1Δ* mutants.

We showed that the 2′-$PO_4$ is required for cleavage, as expression of 'pre-spliced' *HAC1*$^s$ does not lead to cleavage (*Figure 4F and G*). It is possible that the presence of a 2′-$PO_4$ in the context of a composite Ire1 splice site (*i.e.*, formed from two halves of the original 5′- and 3′-splice sites (*Hooks and Griffiths-Jones, 2011*; *Sidrauski and Walter, 1997*)) on *HAC1*$^s$ is recognized by Ire1, but because a 2′-OH is the nucleophile for transesterification by metal-independent Ire1 (*Gonzalez et al., 1999*), the 2′-$PO_4$ inhibits the chemical step of incision. This model also provides a plausible mechanism to explain how Ire1 incises a neighboring, non-canonical site. We propose that a failure of Ire1 to release 2′-phosphorylated *HAC1*$^s$ would enable the active site of a nearby Ire1 molecule—in the context of its activated, oligomeric form (*Korennykh et al., 2009*)—to catalyze site-specific incision at the second, upstream site. Our results also raise the question of why Ire1 would cut incompletely processed *HAC1*$^s$. Here, cleavage of incompletely processed (ligated, but 2′-phosphorylated) *HAC1*$^s$ could be a means to inactivate *HAC1*$^s$ after prolonged stimulation to attenuate the UPR (*Chawla et al., 2011*; *Rubio et al., 2011*).

## Materials and methods

### Yeast strains and plasmids
Yeast strains and sources used in this study are listed in *Table 3*. Plasmids were created as indicated in *Table 3*.

### Cell culture and RNA preparation
Single colonies were inoculated in drop-out media supplemented with relevant amino acids and incubated at 30°C overnight with rotation. Cultures were diluted to an $OD_{600}$ of 0.2 in yeast-extract, peptone, dextrose (YPD) media, and UPR induction was carried out when yeast were growing at mid-log phase with a 2 hr treatment (unless otherwise indicated) with tunicamycin (final concentration of 2.5 μg/mL, Sigma-Aldrich) or DMSO mock treatment. Cells were harvested by centrifugation, and total RNA was isolated by hot acid phenol extraction. For RT-PCR and RT-qPCR experiments, total RNA was treated with TURBO DNase (2 U, Ambion) to degrade contaminating genomic DNA.

### RT-PCR/qPCR
DNase-treated RNA was reverse transcribed with 200 U of SuperScript III reverse transcriptase (Invitrogen) using a gene-specific reverse primer (*Table 2*). Products analyzed on a 1.5% agarose TBE gel, stained with 1x GelRed (Sigma) and imaged with a Bio Rad GelDoc. Densitometry was performed with Bio-Rad Image Analysis software and splicing quantifications were computed and visualized in R using ggplot2 and cowplot R Packages. Quantitative PCR (qPCR) for *KAR2* was also performed on cDNA as generated above, and assayed for *KAR2* and *PGK1* using Sso Advanced Universal SYBR Green Supermix (Bio Rad) and cycled on a Bio Rad C1000 384-well thermal cycler and plate reader. Output $C_t$ values were analyzed in Microsoft Excel and plotted in R using ggplot2 and cowplot R Packages.

**Table 3.** Strain numbers and genotypes.

All strains are background W303 (MATa {leu2-3,112 trp1-1 can1-100 ura3-1 ade2-1 his3-11,15}).

| Strain ID | Genotype | Source |
|---|---|---|
| YJH682 | 'wild-type' (CRY1) | Mingxia Huang |
| YJH632 | xrn1Δ::HygMX | |
| YJH745 | ski2Δ::NatMX | |
| YJH898 | dxo1Δ::KanMX | |
| YJH867 | xrn1Δ::HygMX dxo1Δ::KanMX | |
| YJH829 | tpt1Δ::LEU2 (TPT1 CEN ARS URA3) | Schwer et al., 2004 |
| YJH830 | tpt1Δ::LEU2 (TPT1 CEN ARS URA3) (pAG424-ccdB) | |
| YJH832 | tpt1Δ::LEU2 (TPT1 CEN ARS URA3) (pAG424-10x-tRNA) | |
| YJH834 | tpt1Δ::LEU2 (pAG424-10x-tRNA) | |
| YJH980 | tpt1Δ::LEU2 (pAG424-10x-tRNA) (pAG413-NPr-TPT1) | |
| YJH891 | tpt1Δ::LEU2 (pAG424-10x-tRNA) (pAG413-NPr-tpt1-R138A) | |
| YJH902 | tpt1Δ::LEU2 xrn1Δ::HygMX (pAG424-10x-tRNA) | |
| YJH901 | tpt1Δ::LEU2 ski2Δ::NatMX (pAG424-10x-tRNA) | |
| YJH681 | trl1Δ::KanMX (pRS416-TRL1) | Schwer et al., 2004 |
| YJH708 | trl1Δ::KanMX (pRS416-TRL1) (pAG424) | |
| YJH709 | trl1Δ::KanMX (pRS416-TRL1) (pAG424-10x-tRNA) | |
| YJH835 | trl1Δ::KanMX (pAG424-10x-tRNA) | |
| YJH887 | trl1Δ::KanMX (pAG424-10x-tRNA) (pRS413-TRL1) | |
| YJH811 | trl1Δ::KanMX (pAG424-10x-tRNA) (pRS413-trl1-D425N) | |
| YJH812 | trl1Δ::KanMX xrn1Δ::HygMX (pAG424-10x-tRNA) (pRS413-trl1-D425N) | |
| YJH912 | trl1Δ::KanMX (pAG424-10x-tRNA) (pRS413-trl1-K114A) | |
| YJH913 | trl1Δ::KanMX (pAG424-10x-tRNA) (pRS413-trl1-K114A-D425N) | |
| YJH808 | trl1Δ::KanMX (pRS423-TPI-RtcB) | |
| YJH809 | trl1Δ::KanMX xrn1Δ::HygMX (pRS423-TPI-RtcB) | |
| YJH899 | trl1Δ::KanMX xrn1Δ::HygMX (pAG424-10x-tRNA) | |
| YJH900 | trl1Δ::KanMX ski2Δ::NatMX (pAG424-10x-tRNA) | |
| YJH903 | trl1Δ::KanMX hac1Δ::NatMX (pAG424-10x-tRNA) (pAG413-GPD-HAC1u) | |
| YJH904 | trl1Δ::KanMX hac1Δ::NatMX (pAG424-10x-tRNA) (pAG413-GPD-HAC1s) | |
| YJH920 | hac1Δ::NatMX (pAG424-10x-tRNA) (pAG413-GPD-HAC1u) | |
| YJH921 | hac1Δ::NatMX (pAG424-10x-tRNA) (pAG413-GPD-HAC1s) | |
| YJH923 | tpt1Δ::LEU2 hac1Δ::NatMX (pAG424-10x-tRNA) (pAG413-GPD-HAC1u) | |
| YJH924 | tpt1Δ::LEU2 hac1Δ::NatMX (pAG424-10x-tRNA) (pAG413-GPD-HAC1s) | |

DOI: https://doi.org/10.7554/eLife.42262.012

## Primer extension

Primers specific for HAC1 mRNA and U6 snRNA were PAGE-purified and ethanol precipitated. Oligonucleotide primers were 5'-end-labeled with PNK (Enzymatics) and γ-$^{32}$P-ATP (Perkin Elmer) and purified with Sephadex G-25 spin columns (GE Healthcare resin, Thermo empty columns). Radiolabeled primers and total RNA (15 μg) were heated to 65°C for 5 min and cooled to 42°C. SuperScript III reverse transcriptase (200 U, Invitrogen) was added and reverse transcription reactions were run with a final concentration of 500 μM dNTPs. Primers were extended for 30 min at 42°C, 15 min at 45°C, and 15 min at 50°C. SuperScript III RT was inactivated by heating for 20 min at 75°C. RNA was destroyed with in 10 mM NaOH at 90°C for 3 min and neutralized with HCl. Formamide loading dye was added and products were run on a 8% acrylamide TBE 7M Urea gel. Gels were dried (Bio Rad) and exposed on a phosphor-imager screen and imaged on a Typhoon 9400 (GE Healthcare).

## Northern blotting

Total RNA (3 µg) was electrophoresed on 6% acrylamide TBE 7M urea gels and transferred to nylon membrane (Hybond N+, GE) by electroblotting. Membranes were UV-crosslinked (254 nm, 120 mJ dose), blocked in ULTRAhyb-Oligo Buffer (Ambion), and incubated with 5′-$^{32}$P-labeled oligonucleotide probes (*Table 2*) in ULTRAhyb-Oligo at 42°C for 18 hr. Membranes were washed with 2X SSC/ 0.5% SDS washing buffer two time for 30 min each, exposed on a phosphor-imager storage screen, and imaged on a Typhoon 9400 (GE Healthcare). Membranes were stripped of original probe with three washes in stripping buffer (2% SDS) at 80°C for 30 min per wash. Membranes were re-blocked and probed a second time for the loading control, *SCR1* (*Table 2*).

## Acknowledgements

We thank B Schwer and S Shuman for yeast strains and plasmids, and R Davis and S Jagannathan for comments on the manuscript. We thank Roy Parker for fruitful discussions. This work was supported in part by funding from National Institutes of Health (R35 GM119550 to JH; T32 GM008730 to SP and PC), the RNA Bioscience Initiative at the University of Colorado School of Medicine (PC), and the Victor W Bolie Graduate Scholarship (PC).

## Additional information

### Funding

| Funder | Grant reference number | Author |
| --- | --- | --- |
| University of Colorado | RNA Bioscience Initiative | Patrick D Cherry |
| National Institutes of Health | T32 GM008730 | Patrick D Cherry Sally E Peach |
| University of Colorado | Victor W Bolie and Earleen D Bolie Graduate Scholarship | Patrick D Cherry |
| National Institutes of Health | R35 GM119550 | Jay R Hesselberth |

The funders had no role in study design, data collection and interpretation, or the decision to submit the work for publication.

### Author contributions

Patrick D Cherry, Conceptualization, Data curation, Formal analysis, Validation, Investigation, Visualization, Methodology, Writing—original draft; Sally E Peach, Conceptualization, Methodology, Writing—review and editing; Jay R Hesselberth, Conceptualization, Supervision, Funding acquisition, Visualization, Methodology, Writing—original draft, Project administration, Writing—review and editing

### Author ORCIDs

Patrick D Cherry ⓘ http://orcid.org/0000-0001-6421-2035
Jay R Hesselberth ⓘ https://orcid.org/0000-0002-6299-179X

### Decision letter and Author response

Decision letter https://doi.org/10.7554/eLife.42262.015
Author response https://doi.org/10.7554/eLife.42262.016

## Additional files

### Supplementary files

• Transparent reporting form
DOI: https://doi.org/10.7554/eLife.42262.013

## Data availability

All data generated or analyzed during this study are included in the manuscript and supporting files.

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
