## [Decision Letter]

Thank you for submitting your article "Multiple decay events target *HAC1* mRNA during splicing to regulate the unfolded protein response" for consideration by *eLife*. Your article has been reviewed by two peer reviewers, and the evaluation has been overseen by Timothy Nilsen as the Reviewing Editor and James Manley as the Senior Editor. The reviewers have opted to remain anonymous.

The reviewers have discussed the reviews with one another and the Reviewing Editor has drafted this decision to help you prepare a revised submission.

Summary:

The reviewers were quite positive about the work and thought that it provided fundamental new insight into UPR biology and alternative fates of RNA. That said, both reviewers believed that a few additional straightforward experiments would enhance the impact of the work. The following revisions are required.

Essential revisions:

1) The authors should explicitly do an experiment along the lines of Figure 4F to determine in *tpt1*Δ *hac1*Δ cells if the 5' extended 3' exon or the 3' shortened 5' exon are produced when *HAC1*^u^ or *HAC1*^s^ are expressed. This experiment would unequivocally examine the origin of these fragments.

2) The authors should do a northern as part of Figure 6 to explicitly show that *HAC1*^s^ is substantially increased in *tpt1*Δ *xrn1*Δ and in *tpt1*Δ *ski2*Δ cells, relative to that in *tpt1*Δ cells. It is a major finding, discussed in the Abstract, that this decay is occurring, and the direct comparison by Northern is easy to do.

3) In both examples of the kinase mediated decay pathway (Figure 2 for decay of the 3' exon, and Figure 3 for the decay of the excised intron), the authors have provided convincing evidence that the kinase activity of *Trl1* is required for the accumulation of 3' exon and the intron. However, the authors should also show results from expressing a ligase-dead mutant to demonstrate that ligase activity is not required.

4) In Figure 3, it is not clear to this reader why *trl1*Δ mutants have less accumulated intron than *xrn1*Δ cells, if intron decay is occurring by kinase mediated decay. Would the authors speculate?

---

## [Author Response]

Essential revisions:1) The authors should explicitly do an experiment along the lines of Figure 4F to determine in tpt1Δ hac1Δ cells if the 5' extended 3' exon or the 3' shortened 5' exon are produced when HAC1-u or HAC1-s are expressed. This experiment would unequivocally examine the origin of these fragments.

We generated and verified cells with the *tpt1*∆ *hac1*∆ (*10* *x tRNA*) genotype and transformed them with *HAC1* and *HAC1* expression plasmids. We analyzed HAC1 by northern blot and present the new result in a new panel Figure 4G. We describe the result with the additional sentence: “Furthermore, expression of *HAC1*^s^ in a *tpt1*∆ background also fails to produce additional fragments (Figure 4G), whereas expression of *HAC1* is sufficient in the *tpt1*∆ background to produce the double-band secondary cleavage molecular phenotype.”

2) The authors should do a northern as part of Figure 6 to explicitly show that HAC1-s is substantially increased in tpt1Δ xrn1Δ and in tpt1Δ ski2Δ cells, relative to that in tpt1Δ cells. It is a major finding, discussed in the Abstract, that this decay is occurring, and the direct comparison by Northern is easy to do.

We analyzed both the time 0 and 120 minute samples from the relevant genotypes and blot for *HAC1* 3′-exon, quantifying the splicing percentages in each lane below (Figure 6E).

3) In both examples of the kinase mediated decay pathway (Figure 2 for decay of the 3' exon, and Figure 3 for the decay of the excised intron), the authors have provided convincing evidence that the kinase activity of Trl1 is required for the accumulation of 3' exon and the intron. However, the authors should also show results from expressing a ligase-dead mutant to demonstrate that ligase activity is not required.

We generated and confirmed yeast expression plasmids for the *Trl1* alleles *trl1* -K114A (ligase-dead/adenylylation-dead) (Sawaya et al., 2003) and double-mutant *trl1* -K114A-D425N. We expressed those in a *trl1*∆ (10x *tRNA*) background and performed intron and 3′-exon northern blots on these samples (new Figure 2C and Figure 3I).

In doing so, we add the sentences:

A) “We also tested whether the ligase activity of Trill affected *HAC1* 3′-exon abundance using an adenylyl-transferase/ligase defective allele (*Trl1*-K114A) (Sawaya et al., 2003) and found that additional 3′-exon accumulates compared to wild-type (Figure 2C), indicating that ligation also contributes to processing of free 3′-exon.”

And

B) “Interestingly, expression of the adenylyl-transferase-dead/ligase-dead allele, *trl1* -K114A, also led to accumulation of some free *HAC1* intron (Figure  3I), potentially indicating a role for ligation in the processing of liberated *HAC1*  intron.”

4) In Figure 3, it is not clear to this reader why trl1Δ mutants have less accumulated intron than xrn1Δ cells, if intron decay is occurring by kinase mediated decay. Would the authors speculate?

In fulfilling essential revision 3, we added a blot using *trl1*-K114A, a adenylylation-dead/ligase-dead allele of *Trl1*. This blot showed that both the kinase and adenylylation domains contribute to clearance of liberated *HAC1* intron from the cells. Sentence (2) from above addresses this point.